# Membrane Blue Dual Protects Retinal Pigment Epithelium Cells/Ganglion Cells—Like through Modulation of Mitochondria Function

**DOI:** 10.3390/biomedicines10112854

**Published:** 2022-11-08

**Authors:** Elena Grossini, Sakthipriyan Venkatesan, Micol Alkabes, Caterina Toma, Stefano de Cillà

**Affiliations:** 1Laboratory of Physiology, Department of Translational Medicine, University Eastern Piedmont, 28100 Novara, Italy; 2AGING Project Unit, Department of Translational Medicine, University Eastern Piedmont, 28100 Novara, Italy; 3Eye Clinic, University Hospital Maggiore della Carità, 28100 Novara, Italy; 4Department of Health Sciences, University East Piedmont “A. Avogadro”, 28100 Novara, Italy

**Keywords:** mitochondria, oxidative stress, vital dyes, vitrectomy

## Abstract

Although recent data highlight the greater protective effects exerted by Membrane Blue Dual (MBD), a precise analysis of the mechanisms of action is missing. We examined the effects of MBD with/without polyethylene glycol (PEG) on both human retinal pigment epithelial cells (ARPE-19) and retinal ganglion cells-like (RGC-5) cultured in the presence/absence of ultraviolet B (UVB) treatment on mitochondria function, oxidants, and apoptosis. In ARPE-19/RGC-5 cells either treated or not with UVB, the effects of MBD with/without PEG were evaluated by specific assays for viability, mitochondrial membrane potential and mitochondrial reactive oxygen species (mitoROS) release. Annexin V was used to detect apoptosis, whereas trypan blue and the scratch assay were used for proliferation/migration. In both physiologic conditions and in the presence of UVB, MBD with/without PEG increased cell viability, mitochondrial membrane potential, proliferation and migration in both ARPE-19 and RGC-5 cells. In general, the effects of MBD with PEG were greater than those caused by MBD without PEG. Our results suggest that, in particular, MBD with PEG is a safe and effective dye for vitreoretinal surgery through the modulation of mitochondrial function.

## 1. Introduction

The use of vital dyes to improve visualization of preretinal tissues and membranes has greatly improved vitreomacular surgery in the last 10 years [1,2]. Inadequate visualization of epiretinal membranes (ERMs) and of the inner limiting membrane (ILM) can lead to surgical complications and to irreversible retinal damage as a consequence of retinal pigment epithelium (RPE) and ganglion cells (RGC) damage [1].

The advantages related to the use of the most common dyes [3] include the need for only small quantities and their ease of handling. In addition, however, an optimal dye should stay in the eye for only a short duration of time and should only minimally penetrate the retinal tissues [2]. Despite the aforementioned properties, some studies have reported cytotoxic effects elicited by some vital dyes leading to mitochondrial damage and increased reactive oxygen species (ROS) production [4,5].

Among them, Indocyanine-green (ICG), in particular, has been found to be associated with a risk of damage to the photoreceptors and RPE4 [6,7,8,9] resulting in impaired visual function [10,11]. As for Brilliant blue G^®^ (BBG), reduced cell viability of adult retinal pigment epithelial (ARPE-19) cells in the absence of changes of ROS release were observed only at high concentration and after longer exposure [12]. It is also important to note that, in those cells, GRP78 expression was increased as well, which indicated a kind of protection exerted by BBG in the ARPE-19 cells exposed to light. For other dyes such as Evans blue (EB) and Bromophenol blue (BB), only their interaction with light from an endoillumination source was found to increase the risk of toxicity in ARPE-19 and RGC cells [13].

In the past few years, another vital dye has been introduced into clinical practice, namely Membrane Blue Dual (MBD), which includes both BBG and Trypan blue (TB). By virtue of its increased molecular weight and viscosity properties related to the presence of polyethylene glycol (PEG), it eliminates the need for fluid–air exchange or subretinal injection [14,15].

As for the use of PEG, a recent study on cytotoxicity in RPE cells reported that dye solutions containing 4% PEG showed lower toxicity than those without PEG15. This finding was also confirmed by Awad et al. showing that in ARPE-19 treated with a mixture of BBG, TB and PEG, cell viability was higher than that observed in cells treated with single dye preparations. In that study, no differences were observed regarding the timing of stimulation (5 or 30 min) [16].

In summary, the contradictory reports from all the above studies leave unanswered questions, which also arise from the need to better address the role of vital dyes and their constituents on mitochondria function and oxidants/antioxidants and on cell lines other than those from the RPE.

For the above reasons, in this study we aimed to examine the effects of MBD with/without PEG on both ARPE-19/RGC-5 cells cultured in these physiologic conditions and in the presence of UVB treatment. In these cells, the effects of MBD with/without PEG were examined not only on cell viability and apoptosis but also on mitochondrial membrane potential, mitochondrial ROS release, and proliferation/migration.

## 2. Materials and Methods

### Cell Cultures

Human ARPE-19 cell line was kindly donated by Prof. Matteo Lulli, University of Florence, and was maintained in Dulbecco’s modified Eagle’s medium (DMEM; Sigma, Milan, Italy) supplemented with 10% fetal bovine serum (FBS; Euroclone, Pero, Milan, Italy), 2 mM L-glutamine (Euroclone), and 1% penicillin-streptomycin (P/S; Euroclone) at 37 °C with 5% CO_2_ in incubator.

RGC-5 cells were purchased from ATCC (PTA 6600; American Type Culture Collection, Manassas, VA, USA) and were maintained in Kaighn’s Modification of Ham’s F-12 Medium (F-12K Medium; ATCC), containing 2 mM L-glutamine, 1500 mg/L sodium bicarbonate, and supplemented with 0.1 mg/mL heparin (Sigma), 100 μg/mL endothelial cell growth supplement (ECGS; Sigma), 1% P/S, and 10% FBS.

Membrane Blue Dual with and without PEG was donated by D.O.R.C. (Dutch Ophthalmic Research Center International, Zuidland, The Netherlands).

## 3. Experimental Protocol

### 3.1. Preliminary Phase

In the first part of the study, the ARPE-19 cells and the RGC-5 cells were exposed to UVB light radiation in order to establish the most appropriate dose (dose–response) and timing (time–course) of UVB. UVB radiation was given by an Ultraviolet crosslinker from Amersham Biosciences (Cologno Monzese, Milan, Italy). The irradiation lamp had a wavelength of 302 nm and a filter size of 21 cm × 25 cm. The source of UVB was taken at a 30 cm distance from the cells. For the dose response study, the UVB doses were: 0.0234, 0.117, 0.702, and 1.404 J/cm^2^. The timing of stimulation was 30 min. The rate and timing of radiation were similar to the protocol used during surgery and as previously performed [17,18].

For the time–course study, the ARPE-19 cells and the RGC-5 cells were stimulated with the UVB dose chosen on the basis of the dose–response study, for 5, 30, and 60 min. After each UVB stimulation, the cells were washed with phosphate buffer saline (PBS; Sigma) and conditioned media were replaced. After 24 h, the MTT assay was performed and, based on the results, only one rate of radiation and only one timing of UVB stimulationeach were chosen for the next experiments.

Furthermore, some preliminary experiments were performed on the ARPE-19 cells and the RGC-5 cells in order to examine the effects of MBD with/without PEG on cell viability, at similar doses to the ones used in clinical practice. The concentrations to be used were chosen considering the mean concentration reached by BB and TB in the vitreous after MBD injection: 1.5 × 10^−5^ M and 7.8 × 10^−5^ M, respectively [14].

Starting with this knowledge, the concentrations of MBD with/without PEG were: 1.5 × 10^−6^ M, 7.8 × 10^−6^ M, 3 × 10^−5^ M, 3.9 × 10^−5^ M, 7.8 × 10^−5^ M, 1.6 × 10^−4^ M, and 7.8 × 10^−4^ M.

After 30 min of MBD with/without PEG stimulation, the dye was removed from each well, and a new fresh culture medium was added for 1.5 h [5]. After this time, an analysis of cytotoxicity (MTT assay) was performed on some pools of cells in the absence of UVB or in the presence of UVB at the dose and timing chosen in the grading and time–course study. All the experiments were performed in triplicate and repeated at least 4 times.

### 3.2. Cell Viability

Cell viability was examined in ARPE-19 cells and RGC-5 cells by using the 1% 3-[4, 5-dimethylthiazol-2-yl]-2, 5-diphenyl tetrazolium bromide (MTT; Life Technologies Italia, Monza, Italy) dye, as previously described [19,20,21,22,23,24,25,26]. For the experiments, 10,000 ARPE-19 cells and 10,000 RGC-5 cells for well/insert were plated in 96-well plates and treated with MBD with/without PEG alone (30 min stimulation) or before UVB. After each treatment, the medium was removed, and a fresh culture medium without red phenol or FBS and with 0.5 mg/mL MTT dye was added to the 96-well plate containing the cells that were kept for 2 h at 37 °C in an incubator. Thereafter, the medium was removed, and an MTT solubilization solution (dimethyl sulfoxide; DMSO; Sigma, Milan, Italy) in equal volume to the original culture medium was added and mixed in a gyratory shaker until the complete dissolution of formazan crystals. Cell viability was determined by measuring the absorbance through a spectrometer (VICTOR™ X Multilabel Plate Reader; Perkin Elmer, Milan, Italy) with a wavelength of 570 nm, and cell viability was calculated by setting control cells as 100%.

### 3.3. Extended Phase

In the extended phase of the study, experiments were repeated as described in the preliminary phase except with using only the two concentrations of MBD with/without PEG chosen on the basis of the results observed in the preliminary phase. In particular, we used 7.8 × 10^−4^ M and 3 × 10^−5^ M. We examined the effects of MBD with/without PEG on mitochondrial membrane potential, mitochondrial ROS (mitoROS) release, proliferation, migration, and apoptosis through specific assays. All the experiments were performed in triplicate and repeated at least 4 times.

### 3.4. Mitochondrial Membrane Potential

Mitochondrial membrane potential was measured in the ARPE-19 cells and the RGC-5 cells with a JC-1 assay, as previously described [19,20,27,28] and following the experimental protocol used for the MTT assay. After 30 min stimulation of the ARPE-19 cells and the RGC-5 cells with MBD with/without PEG, as executed for the MTT assay, the dye was removed from each well and a new fresh medium was added for 1.5 h. Thereafter, the medium of the ARPE-19 cells and the RGC-5 cells was removed and the cells were incubated with 5, 51, 6,61-tetrachloro-1, 11, 3,31 tetraethylbenzimidazolyl carbocyanine iodide (JC-1; Cayman Chemical, Ann Arbor, MI, USA) 1X diluted in Assay Buffer 1X for 15 min at 37 °C in an incubator. Further, the cells were washed twice with Assay Buffer 1X and then the mitochondrial membrane potential was evaluated by measuring the red (excitation 550 nm/emission 600 nm) and green (excitation 485 nm/emission 535 nm) fluorescence through a spectrometer (VICTOR™ X Multilabel Plate Reader; Perkin Elmer, Milan, Italy). The ratio of the fluorescent intensity of the J-aggregates to the fluorescent intensity of the monomers was used as an indicator of cell health. The data were normalized versus control cells as 100%.

### 3.5. Mitochondrial ROS (MitoROS) Release

MitoROS production was determined through the Cayman’s Mitochondrial ROS Detection Assay Kit (Cayman Chemical; catalogue number 701,600), as previously performed [28,29]. For the experiments, 10,000 ARPE-19 cells and 10,000 RGC-5 cells/well were plated in 96-well plates in a complete medium, and then the same experimental protocol followed for the MTT and JC-1 methods was used. Briefly, after treatments, the reactions were stopped by removing the culture media and the addition of 120 µL of Cell Based Assay Buffer. After that, the Buffer was aspirated and 100 µL of Mitochondrial ROS Detection Reagent Staining Solution was added in each well, incubated at 37 °C, and protected from light for 20 min. After this time, the Staining Solution was removed and each well was washed with 120 µL of PBS three times. The MitoROS production was measured with an excitation and emission wavelength of 480 and 560 nm, respectively, by using a spectrophotometer (VICTOR™ X Multilabel Plate Reader; Perkin Elmer, Milan, Italy). The data were normalized versus control cells as 100%.

### 3.6. Proliferation Rate

Cell proliferation was evaluated with the TB exclusion method, as previously performed [26]. The ARPE-19 cells and the RGC-5 cells were stimulated as described for all the above assays. At the end of the stimulation with MBD with/without PEG, the cells were detached and 50 µL of cell suspension was diluted 1:2 with TB and mixed by pipetting up and down, and then 10 µL were put in the Burker chamber for cell counting. The percentage of viable cells was calculated by dividing the number of viable cells by the number of total cells and multiplying by 100%.

### 3.7. Migration Rate

Cell migration was evaluated by the in vitro scratch assay, which was executed on 25,000 cells/well (ARPE-19 cells and RGC-5 cells) cultured in 24-well plates, as previously performed [20,24]. In 24-well plates, 25,000 cells/well (ARPE-19 cells and RGC-5 cells) were seeded and cultured in a starvation medium (DMEM without FBS, in order to prevent false positives) to reach a confluent monolayer. Cellular monolayers were then mechanically scratched with a sterile yellow tip along the center of the plate and the cell debris was removed by a gentle washing with PBS (Euroclone, Milan, Italy). Some cell samples were treated as described for the JC-1 and MTT assays and incubated at 37 °C in a humidified atmosphere containing 5% CO2. Images of the cell monolayers were taken using an optical microscope (Leica ICC50HD) with a digital camera to evaluate wound closure. Migration was quantified by calculating the area of wound closure at time points T0 (time of wound), T1.5 (1.5 h after wound) by using ImageJ software. For each condition, the percentage of wound closure at T1.5 h vs. T0 was obtained through the formula:% wound closure: [WA0 − WA/WA0] * 100
where WA = wound area and WA0 = original size of the wound area.

### 3.8. Apoptosis Detection

Apoptotic cells were determined by flow cytometry using an Annexin V, FITC Apoptosis Detection Kit (Dojindo laboratories, supplied by NBS Biologicals Ltd., Cambridgeshire UK). For the experiments, 300,000 cells/well (ARPE-19 cells and RGC-5 cells) in 6-well plates were plated and stimulated as described for the previous assays. After treatments, the reactions were stopped by removing the medium and washing with PBS. Cells were detached with trypsin-EDTA; thereafter, an appropriate volume of culture medium was added and the cell suspension was transferred to a tube and centrifuged at 1000 rpm for 3 min. After centrifugation, the supernatant was removed and 1 mL Annexin V Binding Solution was added to make a final concentration of 1 × 10^6^ cells/mL. Thereafter, 100 μL of this cell suspension was transferred to a new tube and 5 μL Annexin V, FITC conjugate and 5 μL propidium iodide solution was added and incubated for 15 min at room temperature with protection from light. Annexin V Binding Solution (400 μL) was added, and the analysis was performed using Attune NxT (Invitrogen by Thermo Fisher Scientific; Rodano, Milan, Italy).

## 4. Statistical Analysis

All data were recorded using the Institution’s database. Statistical analysis was performed by using STATVIEW version 5.0.1 for Microsoft Windows (SAS Institute Inc., Cary, NC, USA). Data were checked for normality before statistical analysis. All the results obtained were examined through one-way ANOVA followed by Bonferroni post hoc tests. All data are presented as means ± standard deviation (SD) of repeated experiments for each experimental protocol. A value of *p* < 0.05 was considered statistically significant.

## 5. Results

### 5.1. Preliminary Phase

In the ARPE-19/RGC-5 cells treated for 30 min with UVB at different doses, we found a reduction of cell viability. Since the greatest effect was observed at 0.117 J/cm^2^, we used this dose for all subsequent experiments (Figure 1A,B). In addition, the time–course study showed that the 30 min UVB stimulation was the proper one, since 5 min stimulation caused a slight effect and 60 min stimulation almost reset viability (Figure 1C,D).

Furthermore, the results of this phase of study showed that MBD with PEG was able to increase the viability of the ARPE-19/RGC-5 cells cultured in physiologic conditions (Figure 2) at concentrations ranging from 7.8 × 10^−4^ M to 3 × 10^−5^ M. As for MBD without PEG, at 7.8 × 10^−4^ M and 1.6 × 10^−4^ M only, it was able to increase the viability of ARPE-19 cells cultured in physiologic conditions (Figure 2A). In the RGC-5 cells, viability was augmented by using the dye at concentrations ranging from 7.8 × 10^−4^ M to 3 × 10^−5^ M instead (Figure 2B).

In addition, MBD with PEG prevented the effects of UVB in both the ARPE-19/RGC-5 cells. Those effects were observed at concentrations of the dye ranging from 7.8 × 10^−4^ M to 3 × 10^−5^ M and from 7.8 × 10^−4^ M to 7.8 × 10^−6^ M, respectively (Figure 3). Regarding MBD without PEG, in the RGC-5 cells only, we could observe protective effects at concentrations ranging from 7.8 × 10^−4^ M to 3 × 10^−5^ M (Figure 3B). On the basis of these results, we have chosen 7.8 × 10^−4^ M and 3 × 10^−5^ M MBD with/without PEG for our next experiments.

### 5.2. Effects of MBD with/without PEG on Oxidative Stress and Mitochondria Function

MBD with/without PEG increased mitochondrial membrane potential in both the ARPE 19/RGC-5 cells (Figure 4 and Figure 5) cultured in physiologic conditions. In addition, they were able to prevent the effects of UVB on both mitochondrial membrane potential and mitoROS release (Figure 4 and Figure 5).

In the ARPE-19 cells, the effects of MBD with PEG were greater than those caused by MBD without PEG in oxidative stress conditions only (Figure 4A,B; Table 1). In contrast, in the RGC-5 cells (cultured in both physiologic and oxidative stress conditions), the effects on mitochondrial membrane potential of MBD with PEG were greater than those caused by MBD without PEG (Figure 5A,B; Table 2). We also observed similar results with regards to the mitoROS release (Figure 4C,D and Figure 5C,D; Table 1 and Table 2).

### 5.3. Effects of MBD with/without PEG on Proliferation and Migration

In both the ARPE 19/RGC-5 cells, MBD with/without PEG increased proliferation and migration in physiologic conditions and reduced the effects of UVB (Figure 6 and Figure 7). In the case of migration, in both the ARPE-19/RGC-5 cells, the effects were greater when MBD with PEG was used (Figure 7; Table 1 and Table 2).

### 5.4. Effects of MBD with/without PEG on Apoptosis

As shown in Figure 8 and in Table 1 and Table 2, MBD with/without PEG reduced the expression of Annexin V in both the ARPE-19/RGC-5 cells and the effects of MBD with PEG were greater than those caused by MBD without PEG. These results showed anti-apoptotic effects elicited by MBD in the ARPE-19 cells and RGC-5 cells.

## 6. Discussion

The results of this study showed that MBD exerts protective effects on ARPE-19/RGC-5 cells through an improvement of the mitochondrial function.

As a whole, our findings bring additional insights into the use of a particular type of dye used for vitreoretinal surgery, that is MBD. Indeed, previous studies have mainly focused on other kinds of dyes or on the individual components of MBD, which are BBG and TB [30]. Although the conclusions drawn from those studies generally showed cytotoxic effects elicited by most of the dyes [31], in some cases the results were quite contradictory. For example, BBG was reported to be either safe or harmful depending on the concentration, duration of contact, concurrent exposure to light, and kind of solvent used [13,16,31,32,33,34]. Moreover, when using dyes during chromovitrectomy in humans, cytotoxic effects might be also be enhanced by direct light exposure, leading to severe RPE damage with a permanent decrease in visual function [35,36].

Regarding TB, ARPE-19 cells showed no evidence of toxicity, unlike the rat neurosensory retinaR28 cells whose mitochondrial dehydrogenase activity was found to be damaged at the higher doses/light-exposure times [37]. Furthermore, in non-hypoxic conditions, TB increased the adhesive properties of RPE cells to fibronectin, whereas under hypoxia, it suppressed adhesion, reduced proliferation, and increased apoptosis [38].

In our study, we used an experimental protocol that could get as close to the clinical scenario as possible, and in line with previous studies, to examine the response of two retinal cell lines treated with different concentrations of MBD in the presence or absence of an oxidative stress stimulus represented by UVB. Furthermore, we compared the effects of MBD with and without PEG on ARPE-19/RGC-5 cells, which we had chosen since they represent the outer and inner retina, respectively, and could both be affected by dye treatments in patients with ERMs [31]. In addition, among the various factors that can influence the safety of a dye, cell culture models play an important role in deciphering the biological mechanisms involved in evaluating them.

As for the use of RGC-5 cells, although there is controversy in relation to their identity, since some authors believe these cells to be cone photoreceptors while others do not [39], they may still be a useful tool for in vitro screening because they appear to express RGC-typical proteins, such as THY1 (a common marker for retinal ganglion cells) and Opn4 (melanopsin, a photopigment expressed in retinal ganglion cells), as well as neuronal markers such as BRN3 and β tubulin III [40,41]. In addition, Krisnamoorthy et al. [42] corroborated their co-immunoprecipitation data of sigma-1r with L-type voltage-gated calcium channels in both purified RGCs and RGC-5 cells. Finally, in a recent paper, Pan et al. [43] claimed that RGC-5 cells have been established as representative ganglion cells and behave as ganglion cells in the in vitro cultures and, for these reasons, they used those cells for their experiments.

In the preliminary phase focused on cell viability, we used different concentrations of the dyes in order to establish the appropriate doses to be used in the next phase. It should be noted that the chosen concentrations fell within the range of concentrations achievable in the vitreous after MBD injections in clinical practice [14,44].

In addition, we used a 30 min MBD stimulation because it is comparable to the realistically achievable contact that can be observed in the case of interventions performed by even relatively less-experienced surgeons [16]. Moreover, UVB stimulation was used to achieve the oxidative stress stimuli which can arise in the retina as a consequence of endo-illumination and was widely adopted to examine the protective effects elicited by various agents on retinal cells [45,46,47]. The appropriate doses and timings were established in the preliminary phase and based on the information present in the literature [48,49].

As far as cell viability is concerned, only MBD with PEG was able to cause an increase in both the ARPE-19/RGC-5 cells cultured in physiologic conditions at concentrations ranging from 7.8 × 10^−4^ M to 3 × 10^−5^ M. If similar results were obtained with MBD without PEG in the RGC-5 cells, in ARPE-19 cells we could find an increase of cell viability only with the two highest concentrations. Surprisingly, concerning the mitochondrial membrane potential, proliferation and migration rate, both MBD with/without PEG were able to exert positive effects in the ARPE-19/RGC-5 cells cultured in physiologic conditions at 7.8 × 10^−4^ M and 3 × 10^−5^ M.

After UVB stimulation of the ARPE-19/RGC-5 cells, we found a reduction of cell viability, mitochondrial membrane potential, proliferation and migration, but an increase in oxidative stress, as shown by the mitoROS measurement. In addition, apoptosis was strongly potentiated, as evidenced by Annexin V evaluation. All these effects were reduced by the use of MBD with/without PEG. The comparison between the two different dyes, in general, showed greater protective effects on cell viability, mitochondrial membrane potential, mitoROS release, migration, and apoptosis exerted by MBD with PEG. It should also be noted that in the RGC-5 cells, we were able to evidence protection against UVB damages even for lower concentrations of MBD without PEG. This finding is of particular relevance since RGC cells are considered to play a central role in the onset of many ocular neurodegenerative diseases [50]. As a whole, our findings are comparable with previous ones [31] and would support the use of MBD in clinics. As for PEG, the present study could also confirm previous data obtained from RPE and may add information about its safety in patients.

Certainly, the fact that MBD may act as an antioxidant agent, as documented by the reduction of mitoROS release, would represent a new acquisition and could have important clinical relapses, which also applies to data about mitochondrial membrane potential. It should be noted that mitochondrial damage is considered as one of the earliest events, which can lead to apoptosis in response to any cellular stress [51]. Hence, the oxidative damage linked to the alteration of the outer mitochondrial membrane permeabilization causes the release of cytochrome c in the cytosol, which triggers apoptosis through the activation of caspases [52]. In light of this process, the reduction of apoptosis that we observed in the ARPE-19/RGC-5 cells treated with MBD could be attributable to the maintenance of mitochondrial function.

In both RPE and RGCs, oxidative–stress–induced mitochondrial dysfunction is widely considered as common root for the onset of degenerative retinal diseases [53,54,55,56,57]. Thus, the maintenance of mitochondrial function plays a central role not only in regulating the physiological processes within retinal cells but also in counteracting the onset of retinal damages. Due to this, retinal cell mitochondria have been proposed as therapeutic targets for the management of various clinical conditions [58]. In particular, nutraceutical antioxidants have been proven to be effective as both ROS scavengers and inhibitors or inducers of the mitochondrial signaling pathways related to adaptive responses to oxidative stress [50]. For all the above reasons, our findings related to maintaining mitochondrial membrane potential and inhibiting oxidative stress by MBD with/without PEG, could be of particular relevance.

## 7. Conclusions

Our study may add new knowledge about the use of MBD and could be relevant for clinical use. Indeed, use of intraocular dyes in ILM peeling is still controversial [56,57]. Although dyes may be helpful for surgeons to remove ILM, serious adverse effects on the inner retinal layers and damage to the neuroretinal elements can cause great concern [58,59]. The results obtained from the ARPE-19/RGC-5 cells show how the beneficial effects of MBD are expressed through the modulation of the mitochondrial function and of the oxidative stress. In particular, the data regarding the prevalence of beneficial effects carried out by the version of the dye supplemented with PEG are of clinical relevance. Hence, the greatest advantage of using this “heavy” MBD is the improvement of the intraoperative identification of both ERM and ILM discriminating them from surrounding intraocular structures. Due to its increased molecular weight and viscosity properties given by PEG, it eliminates the need for fluid–air exchange, injection of PFCL or subretinal injection, with significant clinical implications.

One limitation of this study could be represented by the use of RGC-5 cells; however, the data we have obtained are corroborated by the knowledge that these cells are widely considered as a valid model for studying physiological and pathophysiological processes in retinal cells [60,61].

## Figures and Tables

**Figure 1 biomedicines-10-02854-f001:**
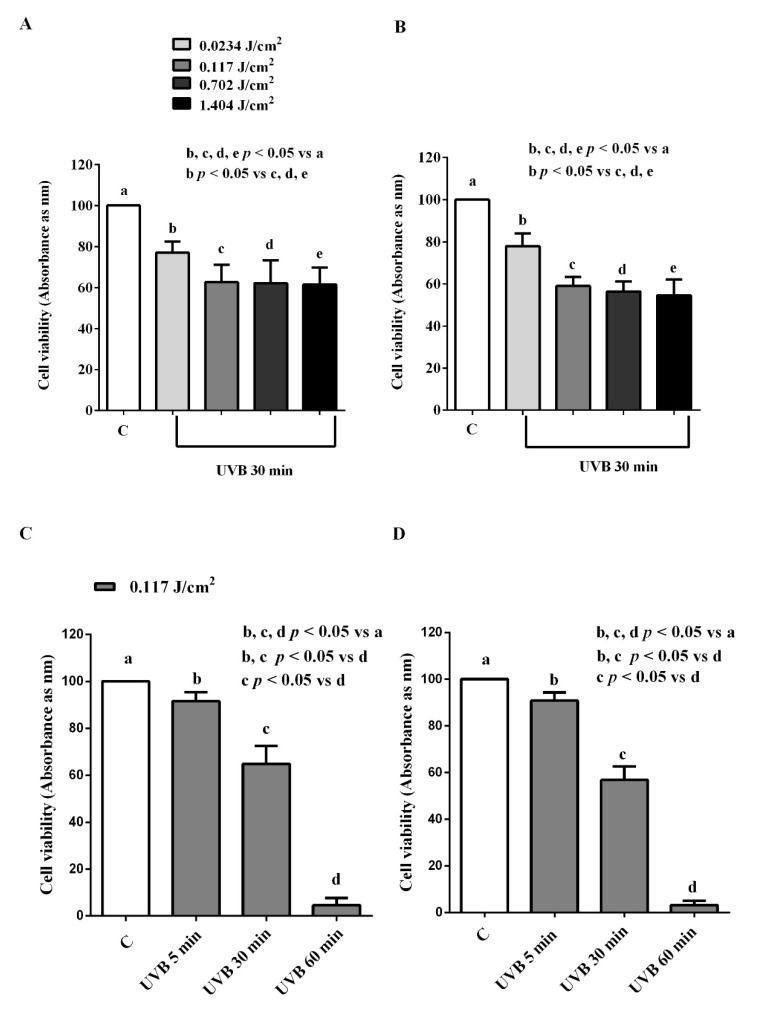
Dose–response (**A**,**B**) and time–course (**C**,**D**) effects of UVB on cell viability of ARPE-19 cells (**A**,**C**) and RGC-5 cells (**B**,**D**). The results are means ± SD of repeated experiments. C: control cells (non-treated).

**Figure 2 biomedicines-10-02854-f002:**
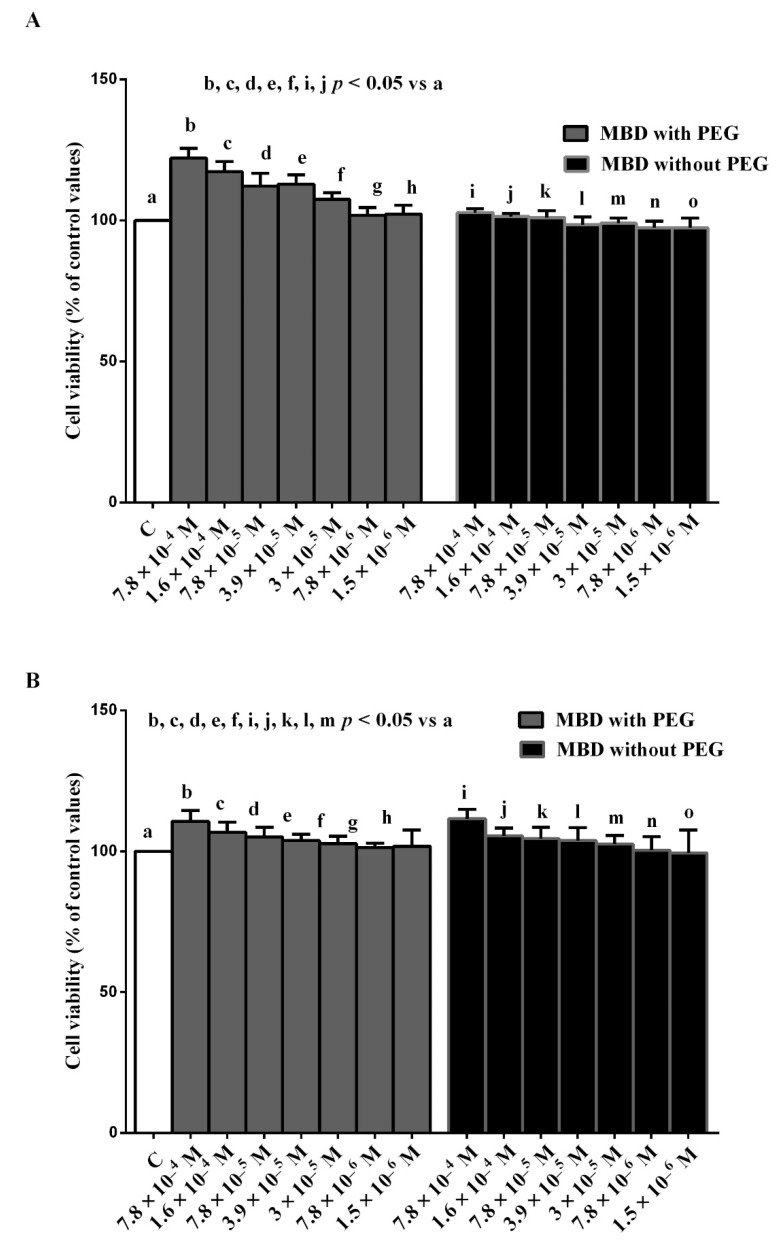
Dose–response effects of 30 min MBD with/without PEG on ARPE-19 cells (**A**) and RGC-5 cells (**B**). The results are means ± SD of repeated experiments. C: control cells (non-treated).

**Figure 3 biomedicines-10-02854-f003:**
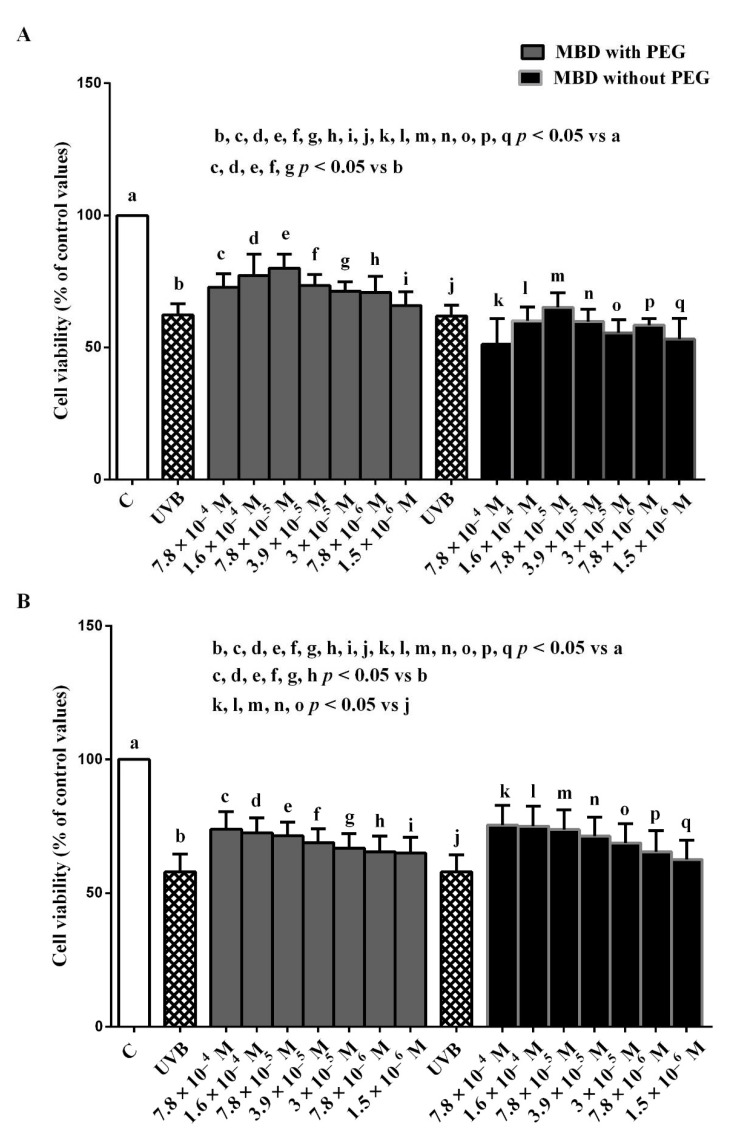
Dose–response effects of 30 min MBD with/without PEG on ARPE-19 cells (**A**) and RGC-5 cells (**B**) in the presence of UVB (0.117 J/cm^2^ for 30 min). The results are means ± SD of repeated experiments. C: control cells (non-treated).

**Figure 4 biomedicines-10-02854-f004:**
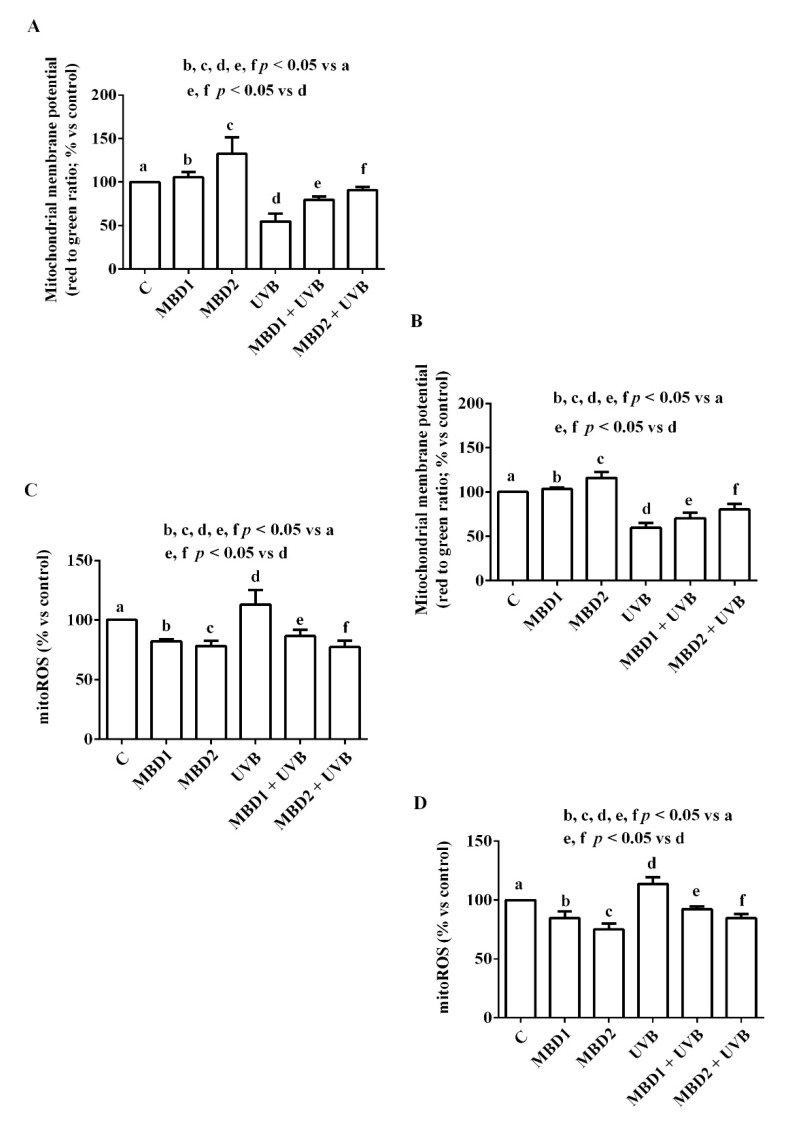
Effects of MBD with (**A**,**C**) and without PEG (**B**,**D**) on mitochondrial membrane potential (**A**,**B**) and mitochondrial reactive oxygen species (mitoROS) release (**C**,**D**) in ARPE-19 cells. MBD1: MBD at 3 × 10^−5^ M. MBD2: MBD at 7.8 × 10^−4^ M. UVB at 0.117 J/cm^2^ (for 30 min). The results are means ± SD of repeated experiments. C: control cells (non-treated).

**Figure 5 biomedicines-10-02854-f005:**
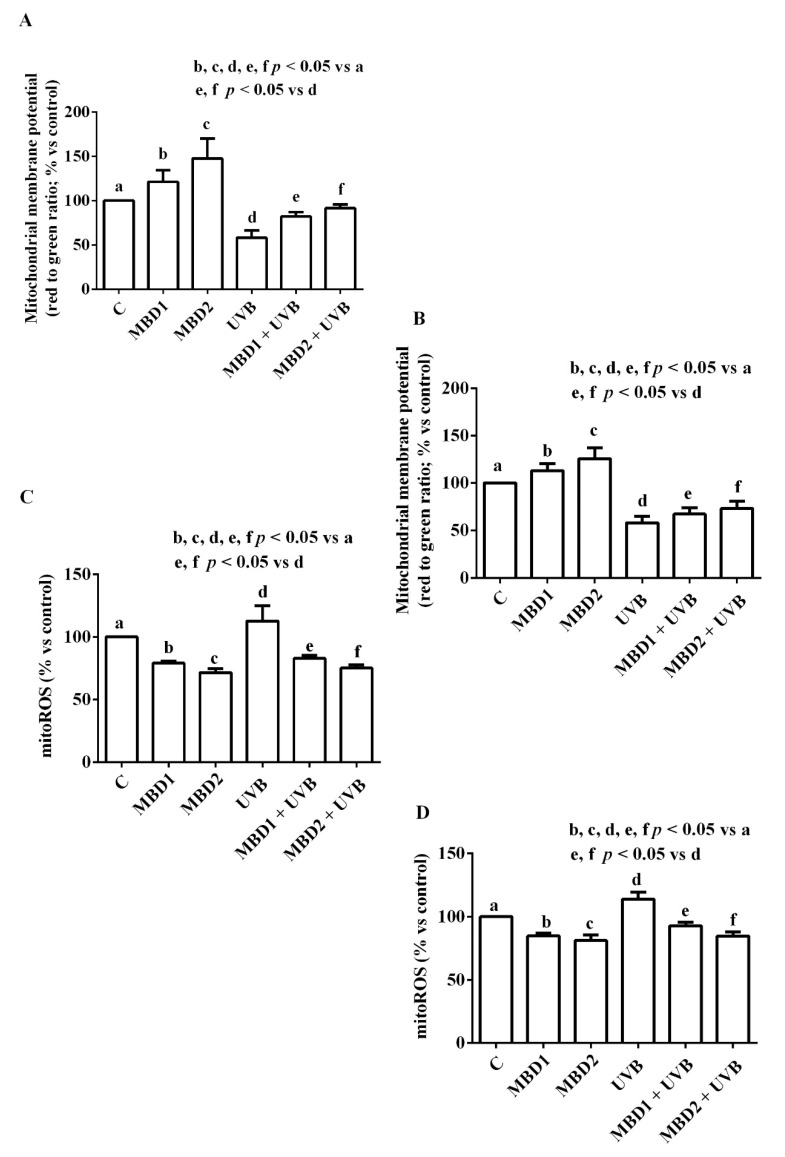
Effects of MBD with (**A**,**C**) and without PEG (**B**,**D**) on mitochondrial membrane potential (**A**,**B**) and mitochondrial reactive oxygen species (mitoROS) release (**C**,**D**) in RGC-5 cells. MBD1: MBD at 3 × 10^−5^ M. MBD2: MBD at 7.8 × 10^−4^ M. UVB at 0.117 J/cm^2^ (for 30 min). The results are means ± SD of repeated experiments. C: control cells (non-treated).

**Figure 6 biomedicines-10-02854-f006:**
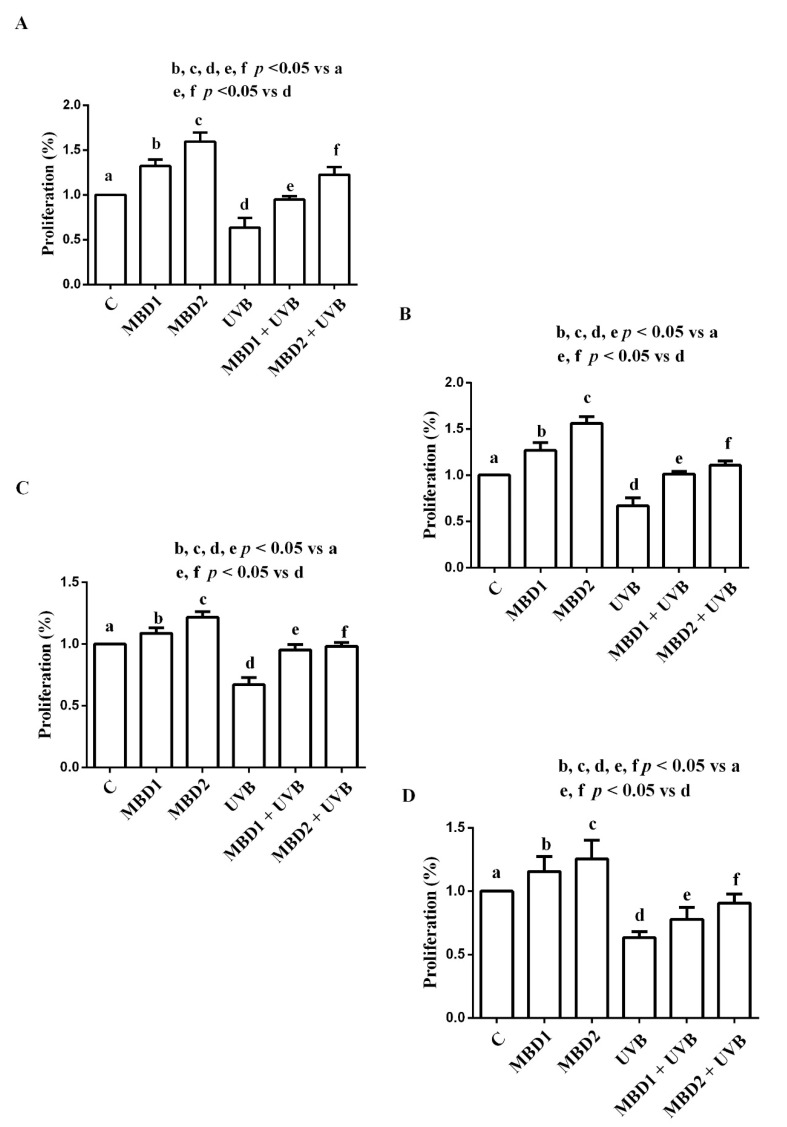
Effects of MBD with (**A**,**C**) and without PEG (**B**,**D**) on proliferation in ARPE-19 cells (**A**,**B**) and RGC-5 cells (**C**,**D**). MBD1: MBD at 3 × 10^−5^ M. MBD2: MBD at 7.8 × 10^−4^ M. UVB at 0.117 J/cm^2^ (for 30 min). The results are means ± SD of repeated experiments. C: control cells (non-treated).

**Figure 7 biomedicines-10-02854-f007:**
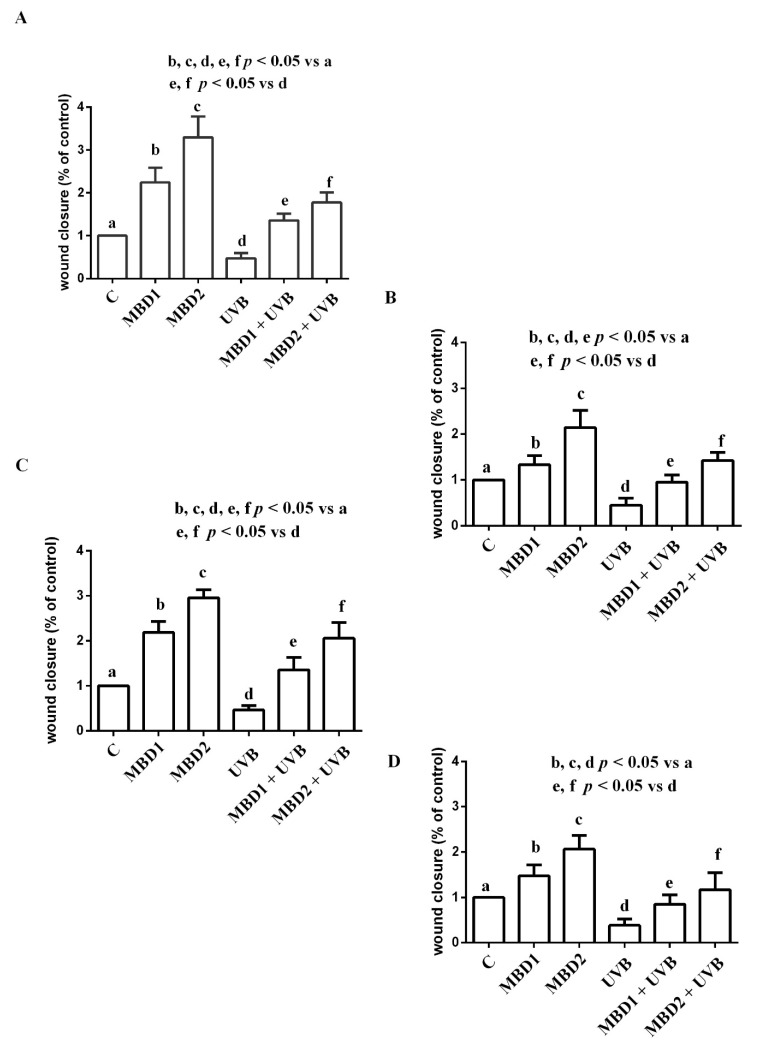
Effects of MBD with (**A**,**C**) and without PEG (**B**,**D**) on migration in ARPE-19 cells (**A**,**B**) and RGC-5 cells (**C**,**D**). MBD1: MBD at 3 × 10^−5^ M. MBD2: MBD at 7.8 × 10^−4^ M. UVB at 0.117 J/cm^2^ (for 30 min). The results are means ± SD of repeated experiments. C: control cells (non-treated).

**Figure 8 biomedicines-10-02854-f008:**
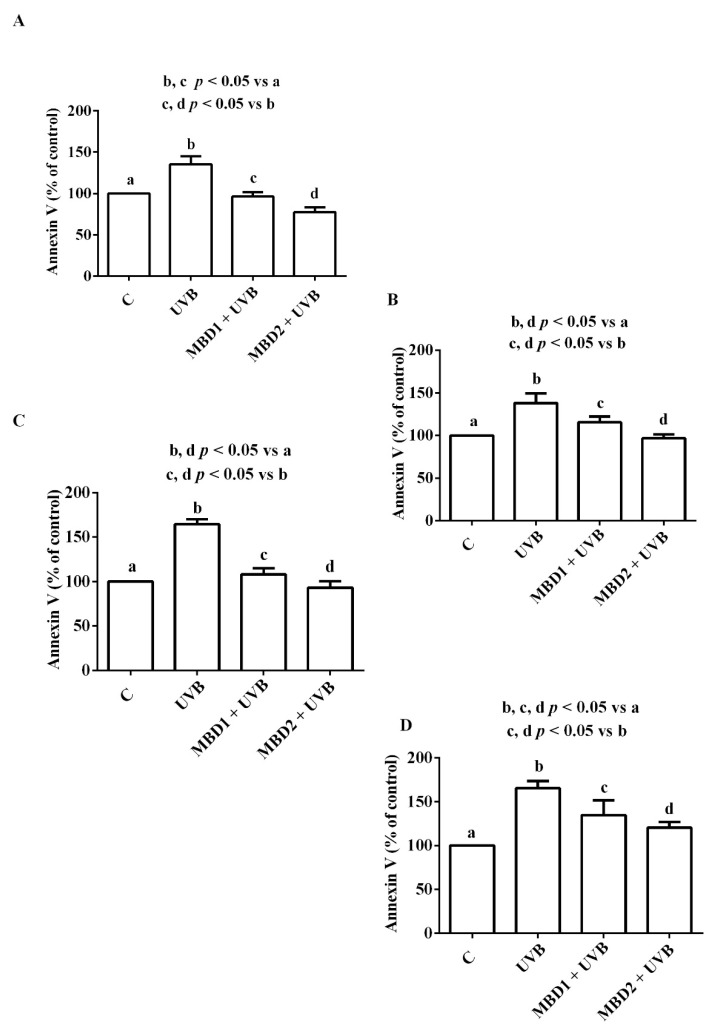
Effects of MBD with (**A**,**C**) and without PEG (**B**,**D**) on apoptosis in ARPE-19 cells (**A**,**B**) and RGC-5 cells (**C**,**D**). MBD1: MBD at 3 × 10^−5^ M. MBD2: MBD at 7.8 × 10^−4^ M. UVB at 0.117 J/cm^2^ for 30 min). The results are means ± SD of repeated experiments. C: control cells (non-treated).

**Table 1 biomedicines-10-02854-t001:** Effects of MBD with/without PEG on ARPE-19 cells.

		With PEG					Without PEG				
	C	MBD1	MBD2	UVB	MBD1+ UVB	MBD2 + UVB	MBD1	MBD2	UVB	MBD1 + UVB	MBD2 + UVB
**Ψ** **(%)**	100	105.6± 5.9	132.4 ± 18.8	54.8 ± 8.7	79.6 ± 3.8	90.8 ± 3.7	103.4 ± 1.7	116 ± 6.5	59.83 ± 5.4	70.4 ± 6.5 **	80.4 ± 6.4 *
**MitoROS (%)**	100	78 ± 4.5	82.17 ± 1.6	113 ± 12.1	77.2 ± 5.6	86.6 ± 5	75 ± 5.2	84.67 ± 5.5	113.8 ± 5.6	92.3 ± 2.2 **	84.7 ± 3.4 *
**Proliferation (%)**	1	1.32 ± 0.07	1.59 ± 0.09	0.634 ± 0.1	0.95 ± 0.03	1.22 ± 0.08	1.27 ± 0.08	1.56 ± 0.07	0.67 ± 0.08	1.01 ± 0.02	1.1 ± 0.04
**Migration (%)**	1	2.24 ± 0.34	3.3 ± 0.48	0.47 ± 0.12	1.36 ± 0.15	1.78 ± 0.23	1.33± 0.2 ^a^	2.14 ± 0.37 ^b^	0.48 ± 0.15	0.95 ± 0.15 **	1.42 ± 0.17 *
**Annexin V (%)**	100			135.6 ± 9.45	96.2 ± 5.76	77.4 ± 6.22			138 ± 11.42	115.6 ± 6.58 **	96.8 ± 4.43 *

MBD: Membrane Blue Dual. MBD1: MBD at 3 × 10^−5^ M. MBD2: MBD at 7.8 × 10^−4^ M. PEG: polyethylene glycol. UVB at 0.117 J/cm^2^ (for 30 min). Ψ: mitochondrial membrane potential. MitoROS: mitochondrial reactive oxygen species. The results are means ± SD of repeated experiments. C: control cells (non-treated). ^a^: MBD1 without PEG *p* < 0.05 vs. MBD1 with PEG. ^b^: MBD2 without PEG *p* < 0.05 vs. MBD2 with PEG. *: MBD1 + UVB, without PEG *p* < 0.05 vs. MBD1 + UVB, with PEG. **: MBD2 + UVB, without PEG *p* < 0.05 vs. MBD2 + UVB, with PEG.

**Table 2 biomedicines-10-02854-t002:** Effects of MBD with/without PEG on RGC-5 cells.

		With PEG					Without PEG				
	C	MBD1	MBD2	UVB	MBD1+ UVB	MBD2 + UVB	MBD1	MBD2	UVB	MBD1 + UVB	MBD2 + UVB
**Ψ** **(%)**	100	121.3± 13.3	147.8 ± 22.3	58.3 ± 7.8	82 ± 5.3	91.67 ± 4.13	113 ± 7.4 ^b^	125.6± 11.6 ^a^	57.86± 6.84	67.17 ± 6.61 *	73.17 ± 7.63 *
**MitoROS** **(%)**	100	79 ± 1.78	71.6 ± 3	112.8 ± 11.92	82.8 ± 2.38	75.2 ± 2.68	84.8± 2.1 ^b^	81.14± 4.45 ^a^	113.8 ± 5.63	92.83 ± 2.78 **	84.67 ± 3.26 *
**Proliferation (%)**	1	1.08 ± 0.04	1.22 ± 0.04	0.67 ± 0.06	0.95 ± 0.045	0.98 ± 0.03	1.15 ± 0.11	1.25 ± 0.14	0.63± 0.047	0.78 ± 0.093	0.9 ± 0.073
**Migration** **(%)**	1	2.1 ± 0.23	2.9 ± 0.18	0.46 ± 0.09	1.35 ± 0.27	2.06 ± 0.34	1.47± 0.24 ^b^	2.064 ± 0.3 ^a^	0.38 ± 0.14	0.85 ± 0.2 **	1.17 ± 0.37 *
**Annexin V** **(%)**	100			164.2 ± 5.84	108 ± 6.67	93.2 ± 7.39			165.4 ± 7.95	134.8 ± 16.6 **	120.2 ± 6.53 *

MBD: Membrane Blue Dual. MBD1: MBD at 3 × 10^−5^ M. MBD2: MBD at 7.8 × 10^−4^ M. PEG: polyethylene glycol. UVB at 0.117 J/cm^2^ (for 30 min). Ψ: mitochondrial membrane potential. MitoROS: mitochondrial reactive oxygen species. The results are means ± SD of repeated experiments. C: control cells (non-treated). ^a^: MBD1 without PEG *p* < 0.05 vs. MBD1 with PEG. ^b^: MBD2 without PEG *p* < 0.05 vs. MBD2 with PEG. *: MBD1 + UVB, without PEG *p* < 0.05 vs. MBD1 + UVB, with PEG. **: MBD2 + UVB, without PEG *p* < 0.05 vs. MBD2 + UVB, with PEG.

## Data Availability

The data presented in this study are available on request from the corresponding author.

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
