# Peer review of "Membrane Blue Dual Protects Retinal Pigment Epithelium Cells/Ganglion Cells—Like through Modulation of Mitochondria Function"

_biomedicines, 2022, doi:10.3390/biomedicines10112854_

Round 1

Reviewer 1 Report

In the article titled " Membrane blue dual protects retinal pigment epithelium cells/ganglion cells through modulation of mitochondria function", the authors aimed to examine the effects of MBD 64 with/without PEG on both ARPE-19/RGC-5 cells cultured in physiologic conditions and in the presence of UVB treatment. The evaluation was made not only on cell viability and apoptosis, but also on mitochondrial membrane potential, mitochondrial ROS release and proliferation/migration.

The introduction provides the necessary information for the further understanding of the subject, including the relevant references in the field and the methodology is accurately described in detail. The experimental protocol gets as close as possible to the clinical scenario. The authors found that MBD exerts protective effects on ARPE19/RGC-5 cells through an improvement of the mitochondrial function, sustaining its use in the clinical practice as a dye for epiretinal membranes and inner limiting membrane as well. This research brings supplementary data on the safety of this relatively new dye for vitreo-retinal surgery, clarifying the contradictory results that have been published on this subject so far. 

The article is written in a scientifically sound style, the experiments are well conducted and described and the results are presented in detail and discussed by reporting them to the most relevant publications in the field. This research has an important impact on the clinical practice with effect on the surgical treatment of various vitreo-retinal disorders.

Author Response

We thank the Reviewer for his positive opinion on our paper and for considering our data important for the clinical practice with effect on the surgical treatment of various vitreo-retinal disor

Reviewer 2 Report

From a first reading, an experimental problem is immediately apparent, making it difficult to evaluate the entire manuscript. One of the cell lines used for the experimental design (RGC-5) are disputed as a rat ganglion cell model since 2013 and this has also led to the withdrawal of previously published work. Moreover, a quick check of the product code given by the authors related to the said cells reveals a discrepancy because that code is associated with another cell line, so the source from which the authors obtained the cells in question also remains to be clarified. In light of this I believe that the work cannot be considered for publication and needs a thorough revision of the experimental design.

Author Response

We agree with the Reviewer that the RGC-5 issue is somewhat debated. As reported by Agarwal N (Agarwal N. RGC-5 Cells. Investigative Ophthalmology & Visual Science December 2013, Vol.54, 7884), the RGC-5 cell line has been widely used since they reproduce several of the original findings and their retinal ganglion cell phenotype. It has been debated whether they may not actually be cone photoreceptors, being similar to the 661W cell line. However, as the two cell types are very different by molecular phenotyping, it is difficult to reconcile this discrepancy. In fact, the same group who published this report, previously used RGC-5 cells for other studies, confirming their nature as retinal ganglion cell. Furthermore, the originator of 661W cells, used RGC-5 cells in several published studies and reported different phenotypic properties between RGC-5 and 661W cell types. Interestingly, the two coauthors of Krishnamoorthy et al. published a paper on sigma-1 receptor (sigma-1r) in purified retinal ganglion cells, in which they corroborated their coimmunoprecipitation data of sigma-1r with L-type volted gated calcium channels in purified RGCs with that of RGC-5 and suggested, “Our co-localization data in purified RGCs is in agreement with the above studies” done with RGC-5 cells”.

That RGC-5 cell line expresses characteristic ganglion cell markers has also been highlighted by other authors. Sun Y et al. states that “The RGC-5 cells express the proteins of Brn3a and βtubulin III, two neuronal cell markers, and the transcripts of Thy1 (a common marker for retinal ganglion cells) and Opn4 (melanopsin, a photopigment expressed in retinal ganglion cells), but not the transcript of Opn1mw (green cone opsins, a marker for cone photoreceptor cells)” (Sun Y, Xue W, Song Z, Huang K, Zheng L. Restoration of Opa1-long isoform inhibits retinal injury-induced neurodegeneration. J Mol Med (Berl). 2016 Mar;94(3):335-46. doi: 10.1007/s00109-015-1359-y).

In the paper by Patil et al (Patil, Singh S, Opere C, Dash A. Sustained-Release Delivery System of a Slow Hydrogen Sulfide Donor, GYY 4137, for Potential Application in Glaucoma AAPS Pharm Sci Tech. 2017 Aug;18(6):2291-2302. doi: 10.1208/s12249-017-0712-6), it is reported that “Although there is controversy with respect to the experimental use of RGC-5 cells and its identity -some believe these cells to be 661 W cells; while others do not, it may still be a useful tool for initial in vitro screening because RGC-5 appears to express RGC-typical proteins such as THY1 and BRN3, as well as neuronal markers.We also confirmed that RGC-5 cells used in this study clearly expressed THY1 and BRN3a proteins through immunocytochemistry”.

Even recently, Pan J et al (Pan J, Liu H, Wu Q, Zhou M. Scopoletin protects retinal ganglion cells 5 from high glucose-induced injury in a cellular model of diabetic retinopathy via ROS-dependent p38 and JNK signaling cascade Cent Eur J Immunol. 2022;47(1):20-29.  doi: 10.5114/ceji.2022.115710. Epub 2022 May 10), say that “The retinal ganglion cell culture RGC-5 has been established as a representative ganglion cell marker, and the cells behave as ganglion cells in an in vitro culture”.

In general, RGC-5 cells are considered a widely used model for studying physiological and pathophysiological processes in retinal cells (Li Y, Chen YM, Sun MM, Guo XD, Wang YC, Zhang ZZ. Inhibition on Apoptosis Induced by Elevated Hydrostatic Pressure in Retinal Ganglion Cell-5 via Laminin Upregulating β1-integrin/Focal Adhesion Kinase/Protein Kinase B Signaling Pathway. Chin Med J (Engl) 2016 Apr 20;129(8):976-83.  doi: 10.4103/0366-6999.179785; Sippl C, Tamm ER. What is the nature of the RGC-5 cell line? Adv Exp Med Biol . 2014;801:145-54.  doi: 10.1007/978-1-4614-3209-8_19).

For the above reasons we believe that the use of RGC-5 cell line in our study may be representative of the effects on an at least ganglion-like cell line. Moreover, the data obtained were compared with those observed from a second retinal cell line, that is ARPE-19.

We have corrected the code of RGC-5 cells.

We agree with the Reviewer that the RGC-5 issue is somewhat debated. As reported by Agarwal N (Agarwal N. RGC-5 Cells. Investigative Ophthalmology & Visual Science December 2013, Vol.54, 7884), the RGC-5 cell line has been widely used since they reproduce several of the original findings and their retinal ganglion cell phenotype. It has been debated whether they may not actually be cone photoreceptors, being similar to the 661W cell line. However, as the two cell types are very different by molecular phenotyping, it is difficult to reconcile this discrepancy. In fact, the same group who published this report, previously used RGC-5 cells for other studies, confirming their nature as retinal ganglion cell. Furthermore, the originator of 661W cells, used RGC-5 cells in several published studies and reported different phenotypic properties between RGC-5 and 661W cell types. Interestingly, the two coauthors of Krishnamoorthy et al. published a paper on sigma-1 receptor (sigma-1r) in purified retinal ganglion cells, in which they corroborated their coimmunoprecipitation data of sigma-1r with L-type volted gated calcium channels in purified RGCs with that of RGC-5 and suggested, “Our co-localization data in purified RGCs is in agreement with the above studies” done with RGC-5 cells”.

That RGC-5 cell line expresses characteristic ganglion cell markers has also been highlighted by other authors. Sun Y et al. states that “The RGC-5 cells express the proteins of Brn3a and βtubulin III, two neuronal cell markers, and the transcripts of Thy1 (a common marker for retinal ganglion cells) and Opn4 (melanopsin, a photopigment expressed in retinal ganglion cells), but not the transcript of Opn1mw (green cone opsins, a marker for cone photoreceptor cells)” (Sun Y, Xue W, Song Z, Huang K, Zheng L. Restoration of Opa1-long isoform inhibits retinal injury-induced neurodegeneration. J Mol Med (Berl). 2016 Mar;94(3):335-46. doi: 10.1007/s00109-015-1359-y).

In the paper by Patil et al (Patil, Singh S, Opere C, Dash A. Sustained-Release Delivery System of a Slow Hydrogen Sulfide Donor, GYY 4137, for Potential Application in Glaucoma AAPS Pharm Sci Tech. 2017 Aug;18(6):2291-2302. doi: 10.1208/s12249-017-0712-6), it is reported that “Although there is controversy with respect to the experimental use of RGC-5 cells and its identity -some believe these cells to be 661 W cells; while others do not, it may still be a useful tool for initial in vitro screening because RGC-5 appears to express RGC-typical proteins such as THY1 and BRN3, as well as neuronal markers.We also confirmed that RGC-5 cells used in this study clearly expressed THY1 and BRN3a proteins through immunocytochemistry”.

Even recently, Pan J et al (Pan J, Liu H, Wu Q, Zhou M. Scopoletin protects retinal ganglion cells 5 from high glucose-induced injury in a cellular model of diabetic retinopathy via ROS-dependent p38 and JNK signaling cascade Cent Eur J Immunol. 2022;47(1):20-29.  doi: 10.5114/ceji.2022.115710. Epub 2022 May 10), say that “The retinal ganglion cell culture RGC-5 has been established as a representative ganglion cell marker, and the cells behave as ganglion cells in an in vitro culture”.

In general, RGC-5 cells are considered a widely used model for studying physiological and pathophysiological processes in retinal cells (Li Y, Chen YM, Sun MM, Guo XD, Wang YC, Zhang ZZ. Inhibition on Apoptosis Induced by Elevated Hydrostatic Pressure in Retinal Ganglion Cell-5 via Laminin Upregulating β1-integrin/Focal Adhesion Kinase/Protein Kinase B Signaling Pathway. Chin Med J (Engl) 2016 Apr 20;129(8):976-83.  doi: 10.4103/0366-6999.179785; Sippl C, Tamm ER. What is the nature of the RGC-5 cell line? Adv Exp Med Biol . 2014;801:145-54.  doi: 10.1007/978-1-4614-3209-8_19).

For the above reasons we believe that the use of RGC-5 cell line in our study may be representative of the effects on an at least ganglion-like cell line. Moreover, the data obtained were compared with those observed from a second retinal cell line, that is ARPE-19.

We have corrected the code of RGC-5 cells.

We agree with the Reviewer that the RGC-5 issue is somewhat debated. As reported by Agarwal N (Agarwal N. RGC-5 Cells. Investigative Ophthalmology & Visual Science December 2013, Vol.54, 7884), the RGC-5 cell line has been widely used since they reproduce several of the original findings and their retinal ganglion cell phenotype. It has been debated whether they may not actually be cone photoreceptors, being similar to the 661W cell line. However, as the two cell types are very different by molecular phenotyping, it is difficult to reconcile this discrepancy. In fact, the same group who published this report, previously used RGC-5 cells for other studies, confirming their nature as retinal ganglion cell. Furthermore, the originator of 661W cells, used RGC-5 cells in several published studies and reported different phenotypic properties between RGC-5 and 661W cell types. Interestingly, the two coauthors of Krishnamoorthy et al. published a paper on sigma-1 receptor (sigma-1r) in purified retinal ganglion cells, in which they corroborated their coimmunoprecipitation data of sigma-1r with L-type volted gated calcium channels in purified RGCs with that of RGC-5 and suggested, “Our co-localization data in purified RGCs is in agreement with the above studies” done with RGC-5 cells”.

That RGC-5 cell line expresses characteristic ganglion cell markers has also been highlighted by other authors. Sun Y et al. states that “The RGC-5 cells express the proteins of Brn3a and βtubulin III, two neuronal cell markers, and the transcripts of Thy1 (a common marker for retinal ganglion cells) and Opn4 (melanopsin, a photopigment expressed in retinal ganglion cells), but not the transcript of Opn1mw (green cone opsins, a marker for cone photoreceptor cells)” (Sun Y, Xue W, Song Z, Huang K, Zheng L. Restoration of Opa1-long isoform inhibits retinal injury-induced neurodegeneration. J Mol Med (Berl). 2016 Mar;94(3):335-46. doi: 10.1007/s00109-015-1359-y).

In the paper by Patil et al (Patil, Singh S, Opere C, Dash A. Sustained-Release Delivery System of a Slow Hydrogen Sulfide Donor, GYY 4137, for Potential Application in Glaucoma AAPS Pharm Sci Tech. 2017 Aug;18(6):2291-2302. doi: 10.1208/s12249-017-0712-6), it is reported that “Although there is controversy with respect to the experimental use of RGC-5 cells and its identity -some believe these cells to be 661 W cells; while others do not, it may still be a useful tool for initial in vitro screening because RGC-5 appears to express RGC-typical proteins such as THY1 and BRN3, as well as neuronal markers.We also confirmed that RGC-5 cells used in this study clearly expressed THY1 and BRN3a proteins through immunocytochemistry”.

Even recently, Pan J et al (Pan J, Liu H, Wu Q, Zhou M. Scopoletin protects retinal ganglion cells 5 from high glucose-induced injury in a cellular model of diabetic retinopathy via ROS-dependent p38 and JNK signaling cascade Cent Eur J Immunol. 2022;47(1):20-29.  doi: 10.5114/ceji.2022.115710. Epub 2022 May 10), say that “The retinal ganglion cell culture RGC-5 has been established as a representative ganglion cell marker, and the cells behave as ganglion cells in an in vitro culture”.

In general, RGC-5 cells are considered a widely used model for studying physiological and pathophysiological processes in retinal cells (Li Y, Chen YM, Sun MM, Guo XD, Wang YC, Zhang ZZ. Inhibition on Apoptosis Induced by Elevated Hydrostatic Pressure in Retinal Ganglion Cell-5 via Laminin Upregulating β1-integrin/Focal Adhesion Kinase/Protein Kinase B Signaling Pathway. Chin Med J (Engl) 2016 Apr 20;129(8):976-83.  doi: 10.4103/0366-6999.179785; Sippl C, Tamm ER. What is the nature of the RGC-5 cell line? Adv Exp Med Biol . 2014;801:145-54.  doi: 10.1007/978-1-4614-3209-8_19).

For the above reasons we believe that the use of RGC-5 cell line in our study may be representative of the effects on an at least ganglion-like cell line. Moreover, the data obtained were compared with those observed from a second retinal cell line, that is ARPE-19.

We have corrected the code of RGC-5 cells.

We agree with the Reviewer that the RGC-5 issue is somewhat debated. As reported by Agarwal N (Agarwal N. RGC-5 Cells. Investigative Ophthalmology & Visual Science December 2013, Vol.54, 7884), the RGC-5 cell line has been widely used since they reproduce several of the original findings and their retinal ganglion cell phenotype. It has been debated whether they may not actually be cone photoreceptors, being similar to the 661W cell line. However, as the two cell types are very different by molecular phenotyping, it is difficult to reconcile this discrepancy. In fact, the same group who published this report, previously used RGC-5 cells for other studies, confirming their nature as retinal ganglion cell. Furthermore, the originator of 661W cells, used RGC-5 cells in several published studies and reported different phenotypic properties between RGC-5 and 661W cell types. Interestingly, the two coauthors of Krishnamoorthy et al. published a paper on sigma-1 receptor (sigma-1r) in purified retinal ganglion cells, in which they corroborated their coimmunoprecipitation data of sigma-1r with L-type volted gated calcium channels in purified RGCs with that of RGC-5 and suggested, “Our co-localization data in purified RGCs is in agreement with the above studies” done with RGC-5 cells”.

That RGC-5 cell line expresses characteristic ganglion cell markers has also been highlighted by other authors. Sun Y et al. states that “The RGC-5 cells express the proteins of Brn3a and βtubulin III, two neuronal cell markers, and the transcripts of Thy1 (a common marker for retinal ganglion cells) and Opn4 (melanopsin, a photopigment expressed in retinal ganglion cells), but not the transcript of Opn1mw (green cone opsins, a marker for cone photoreceptor cells)” (Sun Y, Xue W, Song Z, Huang K, Zheng L. Restoration of Opa1-long isoform inhibits retinal injury-induced neurodegeneration. J Mol Med (Berl). 2016 Mar;94(3):335-46. doi: 10.1007/s00109-015-1359-y).

In the paper by Patil et al (Patil, Singh S, Opere C, Dash A. Sustained-Release Delivery System of a Slow Hydrogen Sulfide Donor, GYY 4137, for Potential Application in Glaucoma AAPS Pharm Sci Tech. 2017 Aug;18(6):2291-2302. doi: 10.1208/s12249-017-0712-6), it is reported that “Although there is controversy with respect to the experimental use of RGC-5 cells and its identity -some believe these cells to be 661 W cells; while others do not, it may still be a useful tool for initial in vitro screening because RGC-5 appears to express RGC-typical proteins such as THY1 and BRN3, as well as neuronal markers.We also confirmed that RGC-5 cells used in this study clearly expressed THY1 and BRN3a proteins through immunocytochemistry”.

Even recently, Pan J et al (Pan J, Liu H, Wu Q, Zhou M. Scopoletin protects retinal ganglion cells 5 from high glucose-induced injury in a cellular model of diabetic retinopathy via ROS-dependent p38 and JNK signaling cascade Cent Eur J Immunol. 2022;47(1):20-29.  doi: 10.5114/ceji.2022.115710. Epub 2022 May 10), say that “The retinal ganglion cell culture RGC-5 has been established as a representative ganglion cell marker, and the cells behave as ganglion cells in an in vitro culture”.

In general, RGC-5 cells are considered a widely used model for studying physiological and pathophysiological processes in retinal cells (Li Y, Chen YM, Sun MM, Guo XD, Wang YC, Zhang ZZ. Inhibition on Apoptosis Induced by Elevated Hydrostatic Pressure in Retinal Ganglion Cell-5 via Laminin Upregulating β1-integrin/Focal Adhesion Kinase/Protein Kinase B Signaling Pathway. Chin Med J (Engl) 2016 Apr 20;129(8):976-83.  doi: 10.4103/0366-6999.179785; Sippl C, Tamm ER. What is the nature of the RGC-5 cell line? Adv Exp Med Biol . 2014;801:145-54.  doi: 10.1007/978-1-4614-3209-8_19).

For the above reasons we believe that the use of RGC-5 cell line in our study may be representative of the effects on an at least ganglion-like cell line. Moreover, the data obtained were compared with those observed from a second retinal cell line, that is ARPE-19.

We have corrected the code of RGC-5 cells.

Round 2

Reviewer 2 Report

After considering the revised version of the manuscript I still have some concerns about the in vitro model of ganglion cells used, but still, I would suggest to the authors to at least mitigate the current title of the manuscript "Membrane blue dual protects retinal pigment epithelium cells/ganglion cells through modulation of mitochondria function" by removing the wording "ganglion cells" and inserting "retinal pigment epithelium cells/ganglion cells - like" or "retinal pigment epithelium cells/neuronal cells".

Author Response

We thank the Reviewer for his suggestion to change the title, which we have performed in the revised version of manuscript.  
